

# Association of rotating night shift work with tooth loss and severe periodontitis among permanent employees in Japan: a cross-sectional study

Yukihiro Sato, Eiji Yoshioka and Yasuaki Saijo

Department of Social Medicine, Asahikawa Medical University, Asahikawa, Hokkaido, Japan

## ABSTRACT

**Background:** The modern 24/7 society demands night shift work, which is a possible risk factor for chronic diseases. This study aimed to examine the associations of rotating night shift work duration with tooth loss and severe periodontitis.

**Methods:** This cross-sectional study used data from a self-administered questionnaire survey conducted among 3,044 permanent employees aged 20–64 years through a Japanese web research company in 2023. The duration of rotating night shift work was assessed using a question from the Nurses' Health Study. Tooth loss was assessed based on self-reported remaining natural teeth count. Severe periodontitis was assessed using a validated screening questionnaire comprising four questions related to gum disease, loose tooth, bone loss, and bleeding gums. We employed linear regression models for tooth loss and Poisson regression models for severe periodontitis, adjusting for demographic, health and work-related variables and socioeconomic status.

**Results:** Among participants included, 10.9% worked in rotating night shifts for 1–5 years, while 11.0% worked in such shifts for ≥6 years. In fully adjusted models, rotating night shift work duration of 1–5 years was associated with tooth loss (beta −0.74, 95% confidence interval (CI) [−1.55 to 0.08]) and severe periodontitis (prevalence ratio 1.80, 95% CI [1.33–2.43]); however, the association with tooth loss was not statistically significant.

**Conclusions:** This study supports that employees who work short-term rotating night shifts may experience poor oral conditions. Further research is needed to determine whether long-term rotating night shift work is associated with deteriorated oral health.

## INTRODUCTION

Oral diseases, such as dental caries, periodontal diseases, and tooth loss, can lead to a decline in work productivity. Dental caries destroy dental hard tissues in the crowns and roots of teeth, causing acute pain (*Selwitz, Ismail & Pitts, 2007*; *Pitts et al., 2017*). Periodontal diseases are chronic inflammatory conditions, which cause pain and

Corresponding author
Yukihiro Sato, ys@epid.work

discomfort (*Pihlstrom, Michalowicz & Johnson, 2005*; *Kinane, Stathopoulou & Papapanou, 2017*). Tooth loss is a complex consequence mainly caused by dental caries and periodontal diseases, which hinder the ability to sleep and communicate (*Sato et al., 2016*; *Haworth et al., 2018*; *Koyama et al., 2018*). Treatment of oral diseases often requires extended clinic visits. Consequently, oral diseases can result in considerable hours lost from work (*Kelekar & Naavaal, 2018*). Moreover, poor periodontal status has been reported to be associated with a potential decline in work performance (*Zaitsu et al., 2020*; *Sato et al., 2023*). Thus, it is crucial to focus on preventing oral diseases among working adults.

Effective prevention of diseases involves focusing on the determinants of health, which encompass the conditions and systems of daily life responsible for health, such as political and economic environments (*Daly et al., 2013*). Work environments play a crucial role as one of these determinants (*Daly et al., 2013*). Accumulated evidence suggests that work environments, such as work-related stress and long working hours, can be detrimental to workers' health (*Lundberg & Cooper, 2010*). However, there is limited evidence on the associations between work environments and oral health. Therefore, there is a need to identify work environment factors potentially related to dental diseases to develop more effective prevention strategies.

The modern 24/7 society demands a diversity of flexible work-hour patterns, and working in night shifts is common in industrialised nations (*IARC, 2020*). In 2012, 21.8% of the employed population of Japan worked in the night shift, a proportion that may continue to increase (*Kubo, 2014*). The proportion of workers engaged in night shift work was 13.3% in the European Union in 2018 (*IARC, 2020*) and 27% in the United States in 2015 (*IARC, 2020*). Night shift work disrupts circadian rhythms, increasing the risk of chronic diseases like diabetes and cardiovascular disease (*Strohmaier et al., 2018*). The possible mechanisms are the induction of oxidative stress, immunosuppression, and chronic inflammation (*Strohmaier et al., 2018*; *IARC, 2020*). Therefore, night shift work can also lead to oral diseases through these mechanisms. In addition, three other potential pathways related to oral diseases are conceivable. The first is the frequency of tooth brushing, which is a fundamental preventive behaviour for dental caries and periodontal diseases. An earlier study indicated that night shift office workers in Japan reported less frequent tooth brushing compared to daytime workers (*Ishizuka et al., 2016*). Therefore, tooth brushing frequency can be a key contributing factor. The second potential pathway is infrequent visits to dental clinics for prevention. *Suzuki et al. (2017)* reported that night work was associated with a disruption in regular dental check-ups. The frequency of visits to dental clinics for prevention might be a factor linking night shift work and poor oral conditions. The third potential pathway is loneliness. Loneliness is a potential risk factor for poor oral health status (*Hajek, Kretzler & König, 2022*). Previous studies have suggested that night shift workers can experience social isolation, which could be a mediator connecting night shift work and chronic diseases through lifestyle and psychosocial factors (*Vetter et al., 2016*; *Cheng & Drake, 2018*). Therefore, loneliness might mediate the association between night shift work and oral diseases.

Our literature search revealed studies related to night shift work and oral diseases. *Han et al. (2013)* reported associations between shift work and periodontitis among 4,597

Korean full-time employees. Shift work, including evening, night, rotating, and irregular schedule work, was associated with periodontitis, with an odds ratio of 1.22. Night workers also had an increased odds ratio of periodontitis, although not statistically significant. White blood cell count partially explained the associations between shift work and periodontitis (*Han et al., 2013*). Two studies reported by *Ishizuka et al. (2016*, *2019)* revealed associations between night shift work and the presence of at least one decayed tooth among 376 male workers and 142 male sales workers. However, *Ghasemi et al. (2022)* indicated null associations of night shift work with decay, missing, filled teeth and >4 mm of periodontal pocket among 180 male workers of a factory. However, these studies had several limitations. First, three of the four studies only focused on male workers and the sample size was relatively small. Second, loneliness was not considered a potential mediator in all the studies. Third, all four studies assessed night shift work status using the current work schedule. *Han et al. (2013)* pointed out the need to consider the impact of accumulated circadian disruption. Therefore, this study aimed to examine cross-sectional associations of rotating night shift work duration with tooth loss and severe periodontitis among 3,044 permanent employees in Japan. In addition, we investigated whether tooth brushing frequency, preventive dental visits, and loneliness mediate the associations.

## MATERIALS AND METHODS

### Ethical approval

This study was approved by the Asahikawa Medical University Research Ethics Committee (No. 22081) in 4 October 2022. The study adhered to the tenets of the Declaration of Helsinki (1983). All participants voluntarily responded to the survey anonymously and provided web-based informed consent before answering the online questionnaire. Participants responded to the questionnaire if they agreed to provide informed consent. Participants had the option to terminate or exit the survey at any time without providing a reason, and not completing the questionnaire was considered non-consent. Participants were incentivised with a credit point that could be used for online shopping and cash conversion.

### Data sources and participants

This cross-sectional study used data from a self-administered questionnaire survey through a Japanese web research company (ASMARQ). The survey took place from 19 to 25 January and from 28 February to 7 March 2023. The inclusion and exclusion criteria are shown in Table 1. We recruited participants using convenience sampling, and the recruitment continued until we reached approximately 3,000 individuals. Out of 142,460 registrants with the Japanese web research company, 29,813 met the inclusion criteria. Among these, 3,132 participants completed the survey (response rate = 10.5%). It is worth noting that a low response rate is relatively common in Japanese Internet surveys (*Tabuchi et al., 2019*).

The following question was included to identify invalid responses: 'Did you answer the previous questions accurately?' Respondents could choose between 'yes, I answered them accurately' and 'no'. Those who selected 'no' were excluded. Of the 3,132 responses, 88

**Table 1 Inclusion and exclusion criteria.**

| Inclusion criteria | Exclusion criteria |
|---|---|
| 1. Aged 20–64 years | 1. Unemployed persons, employers, fixed-term and short-term employees |
| 2. Permanent employees | 2. Non-Japanese speakers |

invalid responses were excluded. Thus, the final analysis included 3,044 permanent employees aged 20–64 years.

## Questionnaire content

### Independent variable: rotating night shift work duration

To gather data on rotating night shift work duration, we used a question from the Nurses' Health Studies (*Vetter et al., 2016*) as follows: 'What is the total number of years during which you worked rotating night shifts (at least 3 nights/month in addition to days or evenings in that month)?' The available response options were 'Never', '1–2 years', '3–5 years', '6–9 years', '10–14 years', '15–19 years', '20–29 years', and '30 years or more'. Subsequently, we categorised them into three levels: 'None', '1–5 years', and '≥6 years or more'.

### Dependent variable: self-reported tooth loss and severe periodontitis

Tooth loss was assessed using self-reported number of teeth. Following prior research (*Ueno et al., 2010, 2018*; *Matsui et al., 2016*), we employed the following question: 'How many natural teeth do you currently have? (excluding wisdom teeth, there are 28 teeth. Including wisdom teeth, there are 32 teeth. Please do not include dental implants, dentures, and dental bridges in your count.)'. Respondents could choose from a range of 0 to 32 teeth.

Severe periodontitis was defined using a validated self-reported questionnaire (*Iwasaki et al., 2021*). This screening questionnaire comprised four oral health questions related to gum disease ('Do you think you might have gum disease?'), loose teeth ('Have you ever had any teeth become loose on their own, without an injury?'), bone loss ('Have you ever been told by a dental professional that you lost bone around your teeth?'), and bleeding gums ('During the past 3 months, have you had bleeding gums?'). Considering the low prevalence of severe periodontitis (6.2% in a previous study) (*Iwasaki et al., 2021*), we defined a score of ≥3 as indicative of severe periodontitis.

### Potential mediator variables: toothbrushing frequency, preventive dental visits, and loneliness

We considered three potential pathways potentially linking night shift work with poor oral health conditions: tooth brushing frequency, preventive dental visits, and loneliness. Toothbrushing frequency was evaluated using a question from the Survey of Dental Diseases ('How many times a day do you brush your teeth?') (*Ministry of Health, Labour and Welfare of Japan, 2016*). We created four categories: 'three times or more a day', 'twice a day', 'once a day', and 'every few days or less'.

The frequency of preventive dental visits was assessed through the following question: 'How often do you receive preventive regular check-ups at a dental clinic? (Please do not include visits for treatment)'. Response options were 'none', 'once every 6 months', 'once a year', 'once every 2 or 3 years'.

Loneliness was measured using the validated Japanese version of the three-item UCLA loneliness scale Version 3: 'How often do you feel that you lack companionship?', 'How often do you feel close to people?', and 'How often do you feel isolated from others?' (*Russell, 1996*; *Arimoto & Tadaka, 2019*). Scores on this scale range from 3 (indicating the lowest level of loneliness) to 12 (indicating the highest level).

### Covariates

Based on previous studies (*Han et al., 2013*; *Ishizuka et al., 2016*, *2019*; *Ghasemi et al., 2022*), we selected the variables below as covariates. Demographic variables were included: age and gender. Work-related variables included the following: occupational category, working hours in the past 7 days, and job title. Occupational categories were determined using the Japan Standard Occupational Classification: administrative and managerial workers, professional and engineering workers, clerical workers, sales workers, service workers, security workers, agriculture, forestry and fishery workers, manufacturing process workers, transport and machine operation workers, construction and mining workers, carrying, cleaning, packaging, and related workers, as well as workers not classified by occupation (*Ministry of Internal Affairs and Communications of Japan, 2009*). Because some job categories were small, they were classified as 'others'. Health-related variables included paper cigarette smoking status, electronic cigarette smoking status, alcohol consumption status, psychological distress according to the Kessler Psychological Distress Scale (K6) (*Ishizuka et al., 2016*; *Vetter et al., 2016*; *Matsui et al., 2016*), medical history of diabetes, and medical history of cardiovascular disease (CVD). In light of socioeconomic status, the following variables were included: marital status, number of persons living together, annual household income, and education level.

### Statistical analysis

Linear regression analysis with a robust error variance was employed to estimate the beta for the number of remaining natural teeth. This method can provide valid estimations even when the dependent variables are skewed (*Schmidt & Finan, 2018*). The beta can be interpreted as the expected differences in the number of teeth. Poisson regression analysis with a robust error variance was employed to estimate prevalence ratios (PRs) for severe periodontitis (*Zou, 2004*). The fully adjusted model included age, gender, occupational category, working hours in the past 7 days, job title, paper cigarette smoking status, electronic cigarette smoking status, alcohol consumption status, psychological distress, medical history of diabetes, medical history of CVD, marital status, number of persons living together, annual household income, and education level. We independently added each potential mediator variable to the fully adjusted model to assess the potential pathway using the difference method (*Judd & Kenny, 1981*). We calculated the percentage change contributed by each potential mediator variable. Statistical significance was set at

two-sided $p < 0.05$ at 95% confidence intervals (CIs). All analyses were performed using the R (ver. 4.3.0; R Foundation for Statistical Computing) for macOS.

## RESULTS

Table 2 presents the basic characteristics, potential mediator variables, night shift work duration, and oral health status of the participants. The mean age was 44.9 years (standard deviation = 10.8). The study population comprised 71.2% men, 28.6% women, and 0.2% others. Among the participants, 10.9% and 11.0% had worked in the night shift for 1–5 years and ≥6 years, respectively. The mean number of natural teeth among the participants was 26.2 (standard deviation = 6.5), and the prevalence of severe periodontitis was 8.4%. The most common occupational category was clerical workers (25.6%), followed by professional workers (22.0%), and administrative and managerial workers (19.3%). Workers with a history of rotating night shift work reported less frequent tooth brushing. The proportion of participants reporting no preventive dental visits was 37.8% among those who did not work night shifts, 38.4% among those who worked rotating night shifts for 1–5 years, and 37.5% among those who worked rotating night shifts for ≥6 years. The mean UCLA loneliness score was 6.3 for non-night shift workers, 6.8 for rotating night shift workers for 1–5 years, and 6.5 for rotating night shift workers for >6 years. Workers who had not experienced rotating night shift work had an average of 26.4 teeth. Workers who worked in night shifts for 1–5 years and ≥6 years had averages of 25.2 and 26.0 teeth, respectively. Workers with experience of rotating night shift work for 1–5 years had the highest prevalence of severe periodontitis (none = 7.5%; 1–5 years = 14.4%; ≥6 years = 8.9%).

Table 3 shows associations of rotating night shift work duration with tooth loss and severe periodontitis. Age and gender-adjusted models showed a statistically significant association between rotating night shift work duration of 1–5 years and tooth loss (beta −1.32, 95% CI [−2.18 to −0.46]). There was no statistically significant association between rotating night shift work duration of ≥6 years and tooth loss (beta −0.52, 95% CI [−1.31 to 0.27]). In fully adjusted models, rotating night shift work duration of 1–5 years was associated with tooth loss, but this was not statistically significant (beta −0.79, 95% CI [−1.61 to 0.03]). Rotating night shift work duration of ≥6 years was not associated with tooth loss (beta −0.02, 95% CI [−0.84 to 0.81]).

In age and gender-adjusted models, compared to workers who had not experienced night shift work, those with rotating night shift work duration of 1–5 years had a statistically significantly increased risk of having severe periodontitis (PR 2.12, 95% CI [1.58–2.84]), but not those with rotating night shift work duration of ≥6 years (PR 1.22, 95% CI [0.84–1.77]). Fully adjusted models indicated only a statistically significant association between rotating night shift work duration of 1–5 years and severe periodontitis (1–5 years: PR 1.80, 95% CI [1.33–2.43]; ≥6 years: PR 1.22, 95% CI [0.83–1.78]). The mediation analyses using the difference method present that the three potential mediator variables did not attenuate the significant association between rotating night shift work duration of 1–5 years and severe periodontitis (toothbrushing frequency: −0.6%; preventive dental visits: −4.2%; loneliness: 0.7%).

**Table 2 Basic statistics of characteristics, potential mediator variables, night shift work duration, and oral health status of participants.**

| | | Total | | Night shift work duration | | | | | |
|---|---|---|---|---|---|---|---|---|---|
| | | | | None | | 1–5 years | | ≥6 years | |
| | | (n = 3,044) | | (n = 2,375, 78.0%) | | (n = 333, 10.9%) | | (n = 336, 11.0%) | |
| **Dependent variables** | | | | | | | | | |
| Number of natural teeth | (Mean, SD) | 26.2 | 6.5 | 26.4 | 6.2 | 25.2 | 7.8 | 26.0 | 7.0 |
| Severe periodontitis | (n, %) | 256 | 8.4 | 178 | 7.5 | 48 | 14.4 | 30 | 8.9 |
| **Mediator variables** | | | | | | | | | |
| Tooth brushing frequency | (n, %) | | | | | | | | |
| Three times or more a day | | 784 | 25.8 | 622 | 26.2 | 76 | 22.8 | 86 | 25.6 |
| Twice a day | | 1,613 | 53.0 | 1,282 | 54.0 | 158 | 47.4 | 173 | 51.5 |
| Once a day | | 592 | 19.4 | 434 | 18.3 | 86 | 25.8 | 72 | 21.4 |
| Every few days or less | | 55 | 1.8 | 37 | 1.6 | 13 | 3.9 | 5 | 1.5 |
| Preventive dental visits | (n, %) | | | | | | | | |
| None | | 1,152 | 37.8 | 898 | 37.8 | 128 | 38.4 | 126 | 37.5 |
| Once every 6 months | | 1,184 | 38.9 | 944 | 39.7 | 120 | 36.0 | 120 | 35.7 |
| Once a year | | 424 | 13.9 | 315 | 13.3 | 56 | 16.8 | 53 | 15.8 |
| Once every 2 or 3 years | | 284 | 9.3 | 218 | 9.2 | 29 | 8.7 | 37 | 11.0 |
| UCLA loneliness scale | (Mean, SD) | 6.4 | 2.4 | 6.3 | 2.4 | 6.8 | 2.3 | 6.5 | 2.5 |
| **Covariates** | | | | | | | | | |
| Age | (n, %) | | | | | | | | |
| 21 to 24 years | | 63 | 2.1 | 48 | 2.0 | 15 | 4.5 | 0 | 0.0 |
| 25 to 29 years | | 252 | 8.3 | 181 | 7.6 | 48 | 14.4 | 23 | 6.8 |
| 30 to 34 years | | 329 | 10.8 | 254 | 10.7 | 39 | 11.7 | 36 | 10.7 |
| 35 to 39 years | | 335 | 11.0 | 248 | 10.4 | 47 | 14.1 | 40 | 11.9 |
| 40 to 44 years | | 381 | 12.5 | 285 | 12.0 | 38 | 11.4 | 58 | 17.3 |
| 45 to 49 years | | 495 | 16.3 | 395 | 16.6 | 41 | 12.3 | 59 | 17.6 |
| 50 to 54 years | | 504 | 16.6 | 392 | 16.5 | 51 | 15.3 | 61 | 18.2 |
| 55 to 59 years | | 439 | 14.4 | 368 | 15.5 | 32 | 9.6 | 39 | 11.6 |
| 60 to 64 years | | 246 | 8.1 | 204 | 8.6 | 22 | 6.6 | 20 | 6.0 |
| Gender | (n, %) | | | | | | | | |
| Men | | 2,166 | 71.2 | 1,638 | 69.0 | 255 | 76.6 | 273 | 81.3 |
| Women | | 872 | 28.6 | 733 | 30.9 | 77 | 23.1 | 62 | 18.5 |
| Others | | 6 | 0.2 | 4 | 0.2 | 1 | 0.3 | 1 | 0.3 |
| Occupational category | (n, %) | | | | | | | | |
| Administrative and managerial workers | | 588 | 19.3 | 489 | 20.6 | 49 | 14.7 | 50 | 14.9 |
| Professional workers | | 670 | 22.0 | 508 | 21.4 | 83 | 24.9 | 79 | 23.5 |
| Clerical workers | | 778 | 25.6 | 715 | 30.1 | 44 | 13.2 | 19 | 5.7 |
| Sales workers | | 235 | 7.7 | 201 | 8.5 | 21 | 6.3 | 13 | 3.9 |
| Service workers | | 246 | 8.1 | 155 | 6.5 | 46 | 13.8 | 45 | 13.4 |
| Manufacturing process workers | | 126 | 4.1 | 74 | 3.1 | 15 | 4.5 | 37 | 11.0 |
| Others | | 401 | 13.2 | 233 | 9.8 | 75 | 22.5 | 93 | 27.7 |
| Working hours in the past 7 days | (n, %) | | | | | | | | |

(Continued)

| | | Total | | Night shift work duration | | | | | |
|---|---|---|---|---|---|---|---|---|---|
| | | | | None | | 1–5 years | | ≥6 years | |
| | | (n = 3,044) | | (n = 2,375, 78.0%) | | (n = 333, 10.9%) | | (n = 336, 11.0%) | |
| 0 to 39 h | | 315 | 10.3 | 233 | 9.8 | 41 | 12.3 | 41 | 12.2 |
| 40 to 44 h | | 888 | 29.2 | 736 | 31.0 | 74 | 22.2 | 78 | 23.2 |
| 44 to 49 h | | 476 | 15.6 | 399 | 16.8 | 39 | 11.7 | 38 | 11.3 |
| 50 to 54 h | | 427 | 14.0 | 320 | 13.5 | 54 | 16.2 | 53 | 15.8 |
| 55 to 59 h | | 248 | 8.1 | 203 | 8.5 | 22 | 6.6 | 23 | 6.8 |
| 60 to 64 h | | 267 | 8.8 | 205 | 8.6 | 33 | 9.9 | 29 | 8.6 |
| 65 to 97 h | | 423 | 13.9 | 279 | 11.7 | 70 | 21.0 | 74 | 22.0 |
| Job title | (n, %) | | | | | | | | |
| None | | 1,524 | 50.1 | 1,191 | 50.1 | 164 | 49.2 | 169 | 50.3 |
| Titled | | 1,520 | 49.9 | 1,184 | 49.9 | 169 | 50.8 | 167 | 49.7 |
| Paper cigarette smoking status | (n, %) | | | | | | | | |
| Never smoked | | 1,412 | 46.4 | 1,163 | 49.0 | 120 | 36.0 | 129 | 38.4 |
| Smoked at least once | | 279 | 9.2 | 195 | 8.2 | 49 | 14.7 | 35 | 10.4 |
| Smoked habitually before | | 660 | 21.7 | 511 | 21.5 | 69 | 20.7 | 80 | 23.8 |
| Smoking occasionally | | 70 | 2.3 | 44 | 1.9 | 14 | 4.2 | 12 | 3.6 |
| Smoking almost every day | | 623 | 20.5 | 462 | 19.5 | 81 | 24.3 | 80 | 23.8 |
| Electronic cigarette smoking status | (n, %) | | | | | | | | |
| Never smoked | | 1,971 | 64.8 | 1,605 | 67.6 | 176 | 52.9 | 190 | 56.5 |
| Smoked at least once | | 290 | 9.5 | 205 | 8.6 | 50 | 15.0 | 35 | 10.4 |
| Smoked habitually before | | 207 | 6.8 | 151 | 6.4 | 30 | 9.0 | 26 | 7.7 |
| Smoking occasionally | | 108 | 3.5 | 73 | 3.1 | 18 | 5.4 | 17 | 5.1 |
| Smoking almost every day | | 468 | 15.4 | 341 | 14.4 | 59 | 17.7 | 68 | 20.2 |
| Alcohol consumption status | (n, %) | | | | | | | | |
| Every day | | 649 | 21.3 | 510 | 21.5 | 70 | 21.0 | 69 | 20.5 |
| 5 to 6 days a week | | 306 | 10.1 | 238 | 10.0 | 33 | 9.9 | 35 | 10.4 |
| 3 to 4 days a week | | 345 | 11.3 | 247 | 10.4 | 46 | 13.8 | 52 | 15.5 |
| 1 to 2 days a week | | 491 | 16.1 | 381 | 16.0 | 60 | 18.0 | 50 | 14.9 |
| A few times a month | | 377 | 12.4 | 295 | 12.4 | 43 | 12.9 | 39 | 11.6 |
| Almost never | | 545 | 17.9 | 440 | 18.5 | 50 | 15.0 | 55 | 16.4 |
| Cannot drink | | 331 | 10.9 | 264 | 11.1 | 31 | 9.3 | 36 | 10.7 |
| Psychological distress | (n, %) | | | | | | | | |
| None | | 1,684 | 55.3 | 1,388 | 58.4 | 119 | 35.7 | 177 | 52.7 |
| Moderate | | 1,013 | 33.3 | 746 | 31.4 | 155 | 46.5 | 112 | 33.3 |
| Severe | | 347 | 11.4 | 241 | 10.1 | 59 | 17.7 | 47 | 14.0 |
| Medical history of diabetes | (n, %) | 121 | 4.0 | 90 | 3.8 | 15 | 4.5 | 16 | 4.8 |
| Medical history of CVD | (n, %) | 49 | 1.6 | 32 | 1.3 | 6 | 1.8 | 11 | 3.3 |
| Marital status | (n, %) | | | | | | | | |
| Married | | 2,022 | 66.4 | 1,579 | 66.5 | 203 | 61.0 | 240 | 71.4 |
| Divorced | | 218 | 7.2 | 169 | 7.1 | 25 | 7.5 | 24 | 7.1 |

| | | Total | | Night shift work duration | | | | | |
|---|---|---|---|---|---|---|---|---|---|
| | | | | None | | 1–5 years | | ≥6 years | |
| | | (*n* = 3,044) | | (*n* = 2,375, 78.0%) | | (*n* = 333, 10.9%) | | (*n* = 336, 11.0%) | |
| Single | | 781 | 25.7 | 613 | 25.8 | 100 | 30.0 | 68 | 20.2 |
| Others | | 23 | 0.8 | 14 | 0.6 | 5 | 1.5 | 4 | 1.2 |
| Number of persons living together | (*n*, %) | | | | | | | | |
| 0 | | 624 | 20.5 | 484 | 20.4 | 83 | 24.9 | 57 | 17.0 |
| 1 | | 725 | 23.8 | 584 | 24.6 | 63 | 18.9 | 78 | 23.2 |
| 2 | | 751 | 24.7 | 591 | 24.9 | 93 | 27.9 | 67 | 19.9 |
| 3 | | 651 | 21.4 | 505 | 21.3 | 60 | 18.0 | 86 | 25.6 |
| ≥4 | | 293 | 9.6 | 211 | 8.9 | 34 | 10.2 | 48 | 14.3 |
| Annual household income | (*n*, %) | | | | | | | | |
| 0 to 3.9 million yen | | 317 | 10.4 | 244 | 10.3 | 44 | 13.2 | 29 | 8.6 |
| 4 to 5.9 million yen | | 625 | 20.5 | 465 | 19.6 | 81 | 24.3 | 79 | 23.5 |
| 6 to 7.9 million yen | | 625 | 20.5 | 486 | 20.5 | 69 | 20.7 | 70 | 20.8 |
| 8 to 9.9 million yen | | 558 | 18.3 | 442 | 18.6 | 51 | 15.3 | 65 | 19.3 |
| 10 to 11.9 million yen | | 365 | 12.0 | 288 | 12.1 | 27 | 8.1 | 50 | 14.9 |
| 12 million yen or more | | 554 | 18.2 | 450 | 18.9 | 61 | 18.3 | 43 | 12.8 |
| Education level | (*n*, %) | | | | | | | | |
| High school or lower | | 509 | 16.7 | 349 | 14.7 | 65 | 19.5 | 95 | 28.3 |
| Professional training college, junior college, and technical college | | 475 | 15.6 | 344 | 14.5 | 54 | 16.2 | 77 | 22.9 |
| University | | 1,796 | 59.0 | 1,470 | 61.9 | 173 | 52.0 | 153 | 45.5 |
| Master's or doctorate's degrees | | 264 | 8.7 | 212 | 8.9 | 41 | 12.3 | 11 | 3.3 |

**Note:**
CVD, cardiovascular disease; SD, standard deviation. Psychological distress was assessed using the Kessler Psychological Distress Scale (K6).

**Table 3 Associations of rotating night shift work duration with tooth loss and severe periodontitis (*n* = 3,044).**

| | Night shift work duration | | | | |
|---|---|---|---|---|---|
| | None | 1–5 years | | ≥6 years | |
| Dependent variable: Number of teeth | | Beta | 95% CI | Beta | 95% CI |
| Age and gender adjusted model | (Reference) | −1.32 | [−2.18 to −0.46] | −0.52 | [−1.31 to 0.27] |
| Fully adjusted model | (Reference) | −0.79 | [−1.61 to 0.03] | −0.02 | [−0.84 to 0.81] |
| Fully adjusted model + Toothbrushing frequency | (Reference) | −0.74 | [−1.55 to 0.08] | −0.04 | [−0.87 to 0.78] |
| % of excess risk explained | | (7.1%) | | (−135.1%) | |
| Fully adjusted model + Preventive dental visits | (Reference) | −0.78 | [−1.59 to 0.03] | 0.01 | [−0.81 to 0.83] |
| % of excess risk explained | | (1.2%) | | (161.3%) | |
| Fully adjusted model + UCLA loneliness scale | (Reference) | −0.80 | [−1.62 to 0.03] | −0.02 | [−0.85 to 0.80] |
| % of excess risk explained | | (−0.2%) | | (−19.7%) | |

(Continued)

| | Night shift work duration | | | | |
| | None | 1–5 years | | ≥6 years | |
| Dependent variable: Number of teeth | | Beta | 95% CI | Beta | 95% CI |
|---|---|---|---|---|---|
| Dependent variable: Severe periodontitis | | PR | 95% CI | PR | 95% CI |
| Age and gender adjusted model | (Reference) | 2.12 | [1.58–2.84] | 1.22 | [0.84–1.77] |
| Fully adjusted model | (Reference) | 1.80 | [1.33–2.43] | 1.22 | [0.83–1.78] |
| Fully adjusted model + Toothbrushing frequency | (Reference) | 1.80 | [1.33–2.43] | 1.21 | [0.83–1.78] |
| % of excess risk explained | | (−0.6%) | | (2.5%) | |
| Fully adjusted model + Preventive dental visits | (Reference) | 1.83 | [1.35–2.48] | 1.23 | [0.84–1.80] |
| % of excess risk explained | | (−4.2%) | | (−5.5%) | |
| Fully adjusted model + UCLA loneliness scale | (Reference) | 1.79 | [1.32–2.42] | 1.21 | [0.83–1.78] |
| % of excess risk explained | | (0.7%) | | (0.9%) | |

Note:
CI, confidence interval; PR, prevalence ratio. Fully adjusted model included age, gender, occupational category, working hours in the past 7 days, job title, paper cigarette smoking status, electronic cigarette smoking status, alcohol consumption status, psychological distress, medical history of diabetes, medical history of CVD, marital status, number of persons living together, annual household income, and education level.

## DISCUSSION

In this study, we analysed cross-sectional data collected from 3,044 permanent employees in Japan. Our findings indicated that rotating night shift work duration of 1–5 years was cross-sectionally associated with an increased risk of tooth loss and severe periodontitis; however, the association with tooth loss was not statistically significant. Rotating night shift work duration of ≥6 years was not statistically significantly associated with tooth loss and severe periodontitis. The three proposed potential pathways did not attenuate the significant association between rotating night shift work duration of 1–5 years and severe periodontitis.

This study had some limitations. Firstly, it is important to note that all variables in this study relied on self-reported data, which introduces the possibility of self-reporting bias. For instance, a previous study indicated that individuals with lower socioeconomic status may tend to report a biased mean number of teeth compared to clinical examination data (*Nascimento et al., 2023*). This information bias could affect the validity of the current results. However, we used validated questionnaires to assess the number of natural teeth (*Ueno et al., 2010*, *2018*; *Matsui et al., 2016*) and periodontitis (*Iwasaki et al., 2021*). Secondly, the current results were obtained from a cross-sectional survey; thus, the temporal relationship of variables was not established, and there is a possibility of reverse causation. Since all the previous studies were cross-sectional (*Han et al., 2013*; *Ishizuka et al., 2016*, *2019*; *Ghasemi et al., 2022*), a cohort study is needed to confirm the validity of the results. Thirdly, it is important to note that participants in our study had a higher socioeconomic status. For example, 18.2% of the participants reported an annual household income of 12 million Yen or more, although the average income reported in a national survey was 6.4 million Yen for worker households in 2022 (*Ministry of Internal Affairs and Communications of Japan, 2022*). This could potentially lead to an

underestimation of the impact on individuals with lower socioeconomic status. Fourthly, the healthy worker effect may have influenced our results (*Li & Sung, 1999*). Unhealthy individuals could have been unable to work due to health conditions. Absent workers and unemployed people who have worked night shifts may not have participated in this survey. Therefore, the impact of night shift work may be underestimated. Our findings indicated that rotating night shift work duration of ≥6 years was not associated with a risk of severe periodontitis, in contrast to rotating night shift work duration of 1–5 years. This result might indicate the presence of the healthy worker effect. Finally, we assessed potential mediator variables (toothbrushing habits, preventive dental visits, and loneliness), but we did not examine biological mechanisms such as oxidative stress, immunosuppression, and chronic inflammation due to the reliance on data solely from a self-administered questionnaire survey. A previous study has indicated that white blood cell count may partially explain the association between shift work and periodontitis (*Han et al., 2013*). Future research should explore these biological mechanisms further.

The duration of night shift work is a more suitable indicator than the current night shift work status because oral diseases are chronic conditions. Night shift work duration considers accumulated circadian disruption. Our results provide new insights into the associations between night shift work and oral diseases. However, in this study, a dose-response relationship was not observed. Earlier studies have explored the existence of a dose-response relationship between night shift work duration and health outcomes (*Wang et al., 2013*; *Torquati et al., 2018*; *Cheng et al., 2019*; *Li et al., 2019*). To examine a dose-response relationship, further investigations should consider the current night shift work status and the duration.

We found an association between night shift work and periodontitis, similar to that of a previous study in Korea (*Han et al., 2013*). Night shift work was associated with an increased risk of tooth loss, but this was not statistically significant. This could be because tooth loss is a multifaceted outcome. In Japan, tooth extraction is mainly attributed to dental caries and periodontal diseases; however, orthodontics and impacted teeth also play a role in tooth loss (*Suzuki et al., 2022*). To gain a more precise understanding of this association in a further studies, it is more appropriate to measure the incidence of tooth loss specifically due to dental caries and periodontitis.

Similar to a previous study (*Han et al., 2013*), our study found no significant role of oral health-related behaviours in linking rotating night shift work duration with oral diseases. In addition, we found that loneliness also did not mediate the association. A study in Korean emphasised the importance of inflammatory markers in connecting night shift work to periodontitis (*Han et al., 2013*). Hence, the previous study (*Han et al., 2013*) and our findings suggest the possibility of a direct pathway as the primary explanation. However, due to the cross-sectional study design, the temporal relationship is not guaranteed. Future cohort studies are needed to further investigate this matter.

A series of studies on night shift work and oral diseases highlights the importance of oral disease prevention among night shift workers. Even though Japan has a universal health coverage system covering dentistry (*Zaitsu, Saito & Kawaguchi, 2018*), previous studies (*Ishizuka et al., 2016*, *2019*) and our study reported the association between night shift

work and oral diseases. Therefore, in countries without dental coverage in their healthcare insurance, this association can be more pronounced. Oral diseases can lead to reduced work productivity (*Zaitsu et al., 2020*; *Sato et al., 2023*) and result in economic burdens (*Righolt et al., 2018*). Night shift work might contribute to producing economic burdens through the development of oral diseases. Employers and managers should recognise night shift work as a potential risk factor for chronic diseases like diabetes, cardiovascular diseases, and oral diseases.

## CONCLUSIONS

This study, involving 3,044 permanent employees in Japan, supports the association between short-term rotating night shift work and poor oral health conditions. The results of the study introduce a new dimension, indicating that not only a history of night shift work but also its duration is associated with dental diseases. Maintaining oral health is crucial for both daytime and night shift workers. However, the association between long-term rotating night shift work and oral conditions remains uncertain. Future research should investigate this association, considering the duration and history of night shift work.

### Funding
This study was supported by a Grant-in-Aid for Early-Career Scientists from the Japan Society for the Promotion of Science KAKENHI (grant number 22K17264). The funders had no role in study design, data collection and analysis, decision to publish, or preparation of the manuscript.

### Grant Disclosures
The following grant information was disclosed by the authors:
Grant-in-Aid for Early-Career Scientists from the Japan Society for the Promotion of Science KAKENHI: 22K17264.

### Competing Interests
The authors declare that they have no competing interests.

### Author Contributions
- Yukihiro Sato conceived and designed the experiments, performed the experiments, analyzed the data, prepared figures and/or tables, authored or reviewed drafts of the article, and approved the final draft.
- Eiji Yoshioka conceived and designed the experiments, analyzed the data, authored or reviewed drafts of the article, and approved the final draft.
- Yasuaki Saijo conceived and designed the experiments, analyzed the data, authored or reviewed drafts of the article, and approved the final draft.

## Human Ethics

The following information was supplied relating to ethical approvals (*i.e.*, approving body and any reference numbers):

This study was approved by the Asahikawa Medical University Research Ethics Committee (No. 22081) in 4 October 2022.

## Data Availability

The raw data is available in the Supplemental File.

## Supplemental Information

Supplemental information for this article can be found online at http://dx.doi.org/10.7717/peerj.17253#supplemental-information.

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
