# Peer review of "Association of rotating night shift work with tooth loss and severe periodontitis among permanent employees in Japan: a cross-sectional study"

_PeerJ, doi:10.7717/peerj.17253_

## Round 0.1 · original submission · Minor Revisions

Thanks for your submission. Please see below some comments to improve the article-

Abstract- the discussion part is confusing, it is mentioned there is no statistical association and this study suggests there is association. Please clear this.

Introduction- Provide a strong rationale using some relevant literature the need for this research.

Methods- IC/EC should be presented in a tabular format.

Conclusion- Should not contain any intext citations.

**Language Note:** The review process has identified that the English language must be improved. PeerJ can provide language editing services - please contact us at copyediting@peerj.com for pricing (be sure to provide your manuscript number and title). Alternatively, you should make your own arrangements to improve the language quality and provide details in your response letter. – PeerJ Staff

·

Basic reporting

this is fine..minor modifications in english suggested in the pdf file.

Experimental design

clear Aims and objectives...

Validity of the findings

Rationale has been explained....
conclusions well stated

Additional comments

can be accepted with minor modifications.

Reviewer 2 ·

Basic reporting

The manuscript was written in clear and good language, However, the literature on this area is rather limited. The basic foundation of the relation between the two variables studied was limited. This work might give some insight into future work focusing on this area.

Experimental design

line 103: " The inclusion criteria were...", are they any exclusion criteria used?

Validity of the findings

no comment

Additional comments

Overall it is a good study that focuses on areas that rarely been highlighted. However, the relationship between the two variables is more complex rather direct relationship. Adding additional information regarding the literature might improve the overall quality of the paper. Thank you

·

Basic reporting

English language could be improved. there are some ambiguities in some parts.
review literature is rather limited. some further studies from more wide regions are required.
more details are included in the main text.

Experimental design

details are included in the main text.

Validity of the findings

details are included in the main text.

Additional comments

details are included in the main text.

---

## Round 0.2 · accepted · Accept

Thanks for making the changes.

·

Basic reporting

the authors have done the minor changes that were requested earlier

Experimental design

the experimental design is robust.

Validity of the findings

they are in line with the experimental design.